# Materials Derived from Olive Pomace as Effective Bioadsorbents for the Process of Removing Total Phenols from Oil Mill Effluents

**DOI:** 10.3390/molecules28114310

**Published:** 2023-05-24

**Authors:** Fatouma Mohamed Abdoul-Latif, Ayoub Ainane, Touria Hachi, Rania Abbi, Meryem Achira, Abdelmjid Abourriche, Mathieu Brulé, Tarik Ainane

**Affiliations:** 1Medicinal Research Institute, Centre d’Etudes et de Recherche de Djibouti, IRM-CERD, Route de l’Aéroport, Haramous B.P. 486, Djibouti City 77101, Djibouti; 2Superior School of Technology of Khenifra, University of Sultan Moulay Slimane, BP 170, Khenifra 54000, Morocco; 3ENSAM Casablanca, University of Hassan II, 150 Bd du Nil, Casablanca 20670, Morocco; 4Laboratory of Biochemical Engineering and Environmental Technology (LBEET), Department of Chemical Engineering, University of Patras, 1 Karatheodori Str., 26504 Patras, Greece

**Keywords:** olive pomace, olive mill effluent, biochar, waste, wastewater, adsorption, phenols, circular economy

## Abstract

This work investigates olive pomace from olive mill factories as an adsorbent for the removal of total phenols from olive mill effluent (OME). This pathway of valorization of olive pomace reduces the environmental impact of OME while providing a sustainable and cost-effective wastewater treatment approach for the olive oil industry. Olive pomace was pretreated with water washing, drying (60 °C) and sieving (<2 mm) to obtain the raw olive pomace (OPR) adsorbent material. Olive pomace biochar (OPB) was obtained via carbonization of OPR at 450 °C in a muffle furnace. The adsorbent materials OPR and OPB were characterized using several basic analyzes (Scanning Electron Microscopy–Energy-Dispersive X-ray SEM/EDX, X-ray Diffraction XRD, thermal analysis DTA and TGA, Fourier transform infrared FTIR and Brunauer, Emmett and Teller surface BET). The materials were subsequently tested in a series of experimental tests to optimize the sorption of polyphenols from OME, investigating the effects of pH and adsorbent dose. Adsorption kinetics showed good correlation with a pseudo-second-order kinetic model as well as Langmuir isotherms. Maximum adsorption capacities amounted to 21.27 mg·g^−1^ for OPR and 66.67 mg·g^−1^ for OPB, respectively. Thermodynamic simulations indicated spontaneous and exothermic reaction. The rates of total phenol removal were within the range of 10–90% following 24 h batch adsorption in OME diluted at 100 mg/L total phenols, with the highest removal rates observed at pH = 10. Furthermore, solvent regeneration with 70% ethanol solution yielded partial regeneration of OPR at 14% and of OPB at 45% following the adsorption, implying a significant rate of recovery of phenols in the solvent. The results of this study suggest that adsorbents derived from olive pomace may be used as economical materials for the treatment and potential capture of total phenols from OME, also suggesting potential further applications for pollutants in industrial wastewaters, which can have significant implications in the field of environmental technologies.

## 1. Introduction

Industrial production of olive oil has expanded in recent years, generating increasing volumes of by-products such as olive mill wastewater (OME) and olive mill solid waste (olive pomace, OP) [1,2], amounting to about 350 kg of pomace and 900 L of OME per ton of freshly harvested olive [3]. OME is particularly toxic due to the presence of long-chain fatty acids (LCFA) forming films on the surface of water bodies, preventing the penetration of light and oxygen, combined with a high concentration of degradable organic matter, along with a slightly acidic nature [4,5,6]. In particular, phenolics are present in high concentrations in OME, contributing to organic load and having phytotoxic, genotoxic and antimicrobial effects, which can hinder natural biodegradation of organic wastes in the environment [7,8]. For efficient remediation of OME, researchers have investigated physical, chemical and biological treatment processes and their combinations, with the main aim of removing phenolic compounds and removing organic matter. In particular, natural biosorbents may allow the capture and recovery of phenolic compounds as high-value biochemicals in a biorefinery approach, particularly as biopesticides for crop treatment and bioconservatives for food conservation [9,10,11].

Adsorption of phenolic compounds is a complex process, which may combine p-interactions, electron donor–acceptor interactions as well as interactions with phenolic compounds already captured on the surface of the adsorbent material [12]. In particular, adsorption may be mediated by free organic acid groups in both protonated and unprotonated forms (RCOOH/RCOO-). Suitable adsorbents may include (1) cost-efficient organic biosorbents derived from organic wastes and by-products, (2) natural mineral adsorbents and (3) synthetic adsorbents such as hydrophilic/hydrophobic polymers. Furthermore, as an alternative to the adsorption process, phenols can also be captured based on the principle of ion exchange by means of ion-exchange resins [13]. Adsorbents can be treated via physical and chemical processing, including pyrolysis, activation, thermal treatment and addition of further compounds [14]. Hence, the possibilities for improvement and optimization of adsorbents and biosorbents are immense, with the main limiting factor being cost efficiency [15].

Biochar is a porous carbonaceous substance obtained from thermochemical organic materials in the absence or presence of oxygen [16], following different methods, including pyrolysis, gasification, torrefaction, hydrothermal carbonization and microwave heating, with varying temperatures and durations [17,18]. A low-cost and environmentally friendly adsorbent, biochar can contribute to carbon capture and storage aiming at reducing CO_2_ concentration in the atmosphere, storing carbon in a stable form during production [19]. Biochar possesses abundant surface functional groups, which make it useful for purifying water, improving soil fertility, producing clean energy and reducing greenhouse gas emissions. Additionally, biochar can be further converted into activated carbon through thermal and chemical reactions or doped with mineral or organic compounds as hybrid/functionalized biochar, enhancing its surface area, porosity, surface charge and yielding higher adsorption efficiencies [20].

In recent years, researchers have explored new approaches to olive waste treatment, including the use of pomace-derived products as well as biochars derived from olive residues and other wastes for the removal of total phenols from OME. Pomace is the solid waste generated during the extraction of olive oil and has high cellulose content. Pomace-derived products, such as biochar and raw material, may be effective materials for the removal of total phenols from OME [21,22,23,24]. Following an alternative pathway, Esteves et al. (2020) [25] applied activated carbons and biochar materials derived from olive mill wastes to develop low-cost catalysts for heterogeneous Fenton-like oxidation of simulated olive mill wastewater, aiming at the reduction of organic carbon content and toxicity.

The present work focuses on the use of raw olive pomace (OPR) as well as biochar obtained through olive pomace pyrolysis (OPB) for the capture and recovery of total phenols, along with chemical and thermal regeneration of the adsorbent.

## 2. Results

### 2.1. Characterization of the Materials

#### 2.1.1. Morphology and Structure of Adsorbent Materials

Surface morphologies of OPR and OPB adsorbents obtained from olive pomace were examined using SEM at a resolution of ×2000 (Figure 1). The obtained images reveal that the surfaces of both materials were heterogeneous, revealing a highly porous surface characterized by cavities and various pore sizes, including macropores, mesopores and micropores. Overall, both structures exhibited promising characteristics for effective adsorption processes, with numerous pores suitable to trap chemical compounds. Nevertheless, the texture of OPB (Biochar) material exhibited a significant difference from that of OPR (raw olive pomace) material regarding the shape and size of the pores.

Micro-analysis using EDX provided some indication of proximate elemental analyses of the adsorbent materials (Table 1). Accordingly, raw olive pomace (OPR) may contain approximately 58.1% carbon and 40.7% oxygen, which is consistent with the cellulosic nature of the material as described in the literature. Conversely, olive pomace biochar (OPB) exhibited a higher percentage of carbon on the surface, amounting to 77.6%.

#### 2.1.2. Crystallinity of Adsorbent Materials

X-ray diffraction analysis is used to investigate the structure of materials at atomic or molecular levels. In the present study, the diffraction patterns obtained from OPR and OPB adsorbent materials may reveal their mainly amorphous nature (Figure 2), as confirmed via the absence of well-defined peaks in diffraction patterns, which implies that both materials may lack long-range order within crystal structures. The diffractogram of OPR peaks slightly around 22.5° and 43°, hinting at the occurrence of crystalline regions within the amorphous matrix of raw olive pomace, owing to the cellulosic nature of the material. On the other hand, the diffraction pattern of OPB (biochar) peaks more sharply around 23°, which may indicate the presence of a carbonaceous reticular plane resulting from the pyrolysis process. In contrast, the single crystalline phase in the diffraction pattern of OPB (biochar) suggests that it is mainly composed of carbon [26,27].

#### 2.1.3. Thermal Analysis

Thermal analyses, such as TGA (thermogravimetric analysis) and DTA (differential temperature analysis), provide valuable information about thermal stability and degradation behavior of materials (Figure 3). In the case of the OPR adsorbent material, TGA analysis revealed that the material has a complex composition, as it consists of cellulose, hemicellulose and lignin fractions. The first weight loss observed in the interval from 25 °C to 150 °C was linked to the elimination of water molecules. The subsequent weight loss, observed from 240 °C, corresponds to the degradation of the cellulosic fraction along with lignin, as confirmed via DTA analysis showing an endothermic peak around 314 °C as an indication for extensive degradation of the material. The important weight loss observed at 240 °C may be attributed to several chemical reactions, including decarboxylation and polymerization of cellulose, hemicellulose and lignin compounds. These reactions result in the breakdown of the material and the release of volatile organic compounds. On the other hand, TGA analysis of OPB adsorbent material showed that the material is mainly composed of carbon. The first weight loss observed in the interval from 50 °C to 450 °C may be related to the removal of remaining water molecules and organic matter. The second weight loss, taking place above 450 °C, may be imputed to the breaking of the chemical bonds of carbon structures, as revealed via the corresponding drop in the DTA curve. These patterns indicate higher thermal stability and higher extent of the carbonization process resulting from the conversion of raw olive pomace into biochar. The highly carbonaceous nature of OPB may be linked to improved characteristics as an adsorbent. Hence, thermal analyses performed in this study highlight, as expected, higher carbon content and thermal stability of OPB (biochar) compared with OPR (raw biomass) [28,29].

#### 2.1.4. Molecular Structure of the Adsorbents

Figure 4 illustrates FTIR spectra of OPR and OPB. Peaks intensities may differ between OPR and OPB, but the groupings are similar. The spectrum of both materials indicates the presence of surface hydroxyl groups, which is confirmed by a broad band observed around 3479 cm^−1^, as a characteristic feature of hydroxyl groups. The broad range of O–H vibration frequencies indicates the occurrence of free hydroxyl groups and bonded O–H bands. C–H stretching of the methyl groups, which are typically present on the lignin structure, is also observed in both materials at 2865 cm^−1^. In addition, the presence of C–O groups in carboxylic and alcoholic groups is identified in both materials at 1035 cm^−1^. This is another common feature of lignin-derived materials, which confirms the presence of these functional groups in both OPR and OPB. Another noteworthy feature of the FTIR spectra is the strong vibration of the C=C bond at around 1660 cm^−1^, suggesting the occurrence of aliphatic and aromatic carbon groups. This is particularly evident in OPB, which is a biochar material, but it is still present in OPR, albeit with a lower intensity [30,31].

#### 2.1.5. Surface and Porosity

Table 2 presents surface and porosity properties of OPR and OPB materials. Both materials possess notable adsorption properties, as demonstrated by their surface characteristics. OPB exhibits 1.7-fold higher pore volume (V_Total_), along with 6.2-fold higher pore surface (S_BET_) compared with OPR, suggesting that OPB may have superior adsorption capabilities. Furthermore, the smaller pore diameter (D_p_) of OPB is slightly above the classification limit for microporous material (<2 nm), hinting at optimized adsorption capacity for small molecules, such as phenols.

### 2.2. Removal of Phenolic Compounds using Pomace-Derived Products

#### 2.2.1. Effect of Adsorption Parameters

The optimization of batch TPC adsorption in pre-diluted OME samples targeted the parameters of pH (range 4–10) and adsorbent dose, while other parameters such as temperature, reaction time, initial dosage of adsorbent and initial TPC concentration were kept constant at T = 298 K (25 °C), t = 24 h, [Adsorbent] = 2 g·L^−1^, [TPC] = 100 ppm (or 0.1 g/L). Highest TPC removal performances were achieved at a pH of 10 for OPR and OPB, amounting to 30.9% and 58.4%, respectively, as shown in Figure 5. In accordance with the maximum removal performance obtained in the optimization experiment, all subsequent experiments were conducted at pH 10. Increased pH levels may result in strengthened electrostatic interactions between polyphenols and adsorbent materials, leading to greater stability of alcohol–phenol bonds and increased adsorption of polyphenols on the surface of the material, which is particularly valid for biochar. Studies have shown that biochar exhibits a higher adsorption capacity for hydroxyl ions, which are more prevalent at higher pH, compared with free ions. This is supported by our experimental findings, obtained while increasing pH from 4 to 10, which indicate that increasing pH may enhance the adsorption capacity of biochar for polyphenols. Hence, optimizing the pH could significantly improve the efficacy of biochar as an adsorbent for polyphenols [32,33].

According to Figure 6, the removal rate of TPC from the two materials increased along with increasing initial weight of added adsorbent. Specifically, the removal rate increased from 11.5% to 38.2% for OPR and from 35.5% to 89.1% for OPB, along with increased adsorbent dosage from 1 to 20 g/L, which may be related to increased availability of adsorption sites [34].

#### 2.2.2. Sorption Kinetics

The study investigated the adsorption kinetics of total phenols in pre-diluted OME via OPR and OPB based on experimental data with contact time varied from 0 to 24 h, while setting initial TPC concentration at 100 ppm and the dosage of adsorbent material at 2 g·L^−1^. The resulting curves of adsorption kinetics for both materials were plotted and analyzed (Figure 7), and the linear form of model equations for both pseudo-first-order and pseudo-second-order models was used to estimate kinetic parameters (Table 3). Compared with the pseudo-first-order model, the pseudo-second-order model yielded a better correlation to experimental data. Correlation coefficients (*R*^2^) of the pseudo-second-order model were 0.902 for OPR and 0.970 for OPB, respectively, indicating a good fit between experimental data and model predictions. One interpretation of the kinetics behavior implies that the adsorption process may be controlled via chemisorption, where phenol molecules are adsorbed via chemical bonds, rather than physical adsorption, which is based on van der Waals forces. Kinetics analysis also reveals that the adsorption of total phenols via OPR and OPB is a slow process, so that a long contact time is required for more complete removal of total phenols from the solution [35].

#### 2.2.3. Adsorption Isotherms

Adsorption isotherms may contribute to a better understanding of interactions between adsorbate (total phenols) and adsorbent (OPR and OPB). The isotherm study conducted on both materials was carried out for an experimental duration of 24 h, and the initial TPC concentration was set to 100 ppm with a pH of 10. Experiments were repeated at different dosages of adsorbent material. The resulting data were used to plot the variation in adsorption capacities (*Q_e_*) as a function of equilibrium concentrations (*C_e_*) (Figure 8).

Experimental data for adsorption isotherms were described according to several models, including Langmuir and Freundlich models. The parameters for each isotherm were derived from linear relationships between model equations. The isotherm model yielding the best fit to experimental data was subsequently selected according to correlation coefficients (*R*^2^).

Based on calculated *R*^2^ values (Table 4), the Langmuir model provided the best fit to experimental values of TPC adsorption via both OPR and OPB materials. This model assumes a monolayer adsorption of the adsorbate onto the adsorbent surface, corresponding to a finite number of adsorption sites with equivalent energies. The values of maximum adsorption capacity (Q_max_) for the Langmuir model amounted to 21.27 mg·g^−1^ for OPR and 66.67 mg·g^−1^ for OPB, respectively, indicating that OPB yielded higher adsorption capacity for total phenols.

The Langmuir isotherm is widely used to characterize the adsorption of solutes onto surfaces. One of the important parameters of the Langmuir isotherm is the separation factor, or equilibrium parameter, *R_L_*, which is an indicator of the type of adsorption that takes place. The *R_L_* value can be calculated as follows [36]:(1)$$R_{L}=\frac{1}{1+K_{L}\cdot C_{0}}$$
where *K_L_* accounts for the Langmuir constant (L·mg^−1^), *C*_0_ for initial concentration (ppm) and *R_L_* for the separation factor. The value of *R_L_* indicates the type of adsorption, with *R_L_* = 0 representing irreversible adsorption, while *R_L_* = 1 represents linear adsorption, and *R_L_* < 1 suggests favorable adsorption, while *R_L_* > 1 indicates unfavorable adsorption [37].

In our study, *R_L_* values were found to be 0.30 for OPR and 0.22 for OPB, which implies that the adsorption process was favorable. This indicates that both OPR and OPB adsorbent materials are capable of efficiently adsorbing total phenols from aqueous solutions.

The Langmuir model equation predicts a monolayer adsorption process, where the adsorbate molecules are adsorbed onto the adsorbent surface and form a single layer. The adsorption process follows the law of mass action, which states that adsorption rate is proportional to adsorbate concentration at the surface of the adsorbent. Furthermore, in the Langmuir model, adsorption is assumed to occur on a surface carrying a finite number of identical sites, and adsorption may only occur when a site is empty. Hence, the adsorption of one molecule may not affect the adsorption of another molecule and lateral interactions between adsorbed molecules may not occur [38].

#### 2.2.4. Thermodynamic Study

Thermodynamic parameters of adsorption processes provide valuable information regarding the feasibility and spontaneity of the reaction. In this study, the Gibbs free energy change (Δ*G*°), enthalpy change (Δ*H*°) and entropy change (Δ*S*°) were estimated based on experimental data obtained at different temperatures. The equilibrium constant (*K* = *K_L_*) was obtained from Langmuir isotherms of different temperatures and applied to estimate thermodynamic parameters (Equations (1), (12)–(14)).

The slope and intercept of the plot of ln *K_L_* versus 1/*T* provided the values of Δ*H*° and Δ*S*° (Table 5). The enthalpy change (Δ*H*°) values were −13.15 and −21.38 KJ·K^−1^·mol^−1^ for OPR and OPB, respectively. The entropy change (Δ*S*°) amounted to −0.054 and −0.091 KJ·K^−1^·mol^−1^ for OPR and OPB, respectively. Standard enthalpy change values (Δ*H*°) were negative, indicating that adsorption may be exothermic. Furthermore, Δ*S*° was also negative, suggesting a decrease in randomness at the solid/solution interface during the adsorption process. Negative values for Δ*H*° and Δ*S*° may indicate that the adsorption of TPC via OPR and OPB is feasible, spontaneous and exothermic, so that both OPR and OPB may be effective adsorbents for TPC removal from aqueous solutions [39].

### 2.3. Effect of Regeneration

The reuse of regenerated OPR and OPB was investigated in batch adsorption experiments with pre-diluted OME with initial TPC concentration set at 100 ppm. Results indicated a decrease in sorption capacities for both materials following thermal or chemical regeneration (Figure 9). Highest TPC removal efficiencies were achieved with biochar (OPB) following thermal regeneration at 200 °C (~40%) as well as chemical regeneration using 70% alcohol (≅41%).

## 3. Discussion

A study was conducted to investigate the use of pomace-derived products for the removal of total phenols from OME. The study focused on two pomace-derived products: raw dried olive pomace (OPR) and biochar (OPB).

Pomace-derived products were characterized as adsorbent materials by means of SEM/EDX, XRD, TGA-DTA and FTIR. According to these analyses, both products displayed porous organic textures, with OPR presumably containing a high share of cellulose, hemicellulose and lignin, while OPB had a less crystalline, more disordered chemical structure consisting mainly of carbon. XRD is widely used for the characterization of biochar, and its main aim is to differentiate materials according to their crystalline or amorphous nature. When comparing raw biomass with biochar, the decrease in crystallinity may reveal the conversion of cellulose into amorphous carbonaceous material achieved through pyrolysis at 450 °C. However, in XRD analyses of activated carbons produced through high-temperature thermal treatment of biochar, crystalline graphitic structures may appear again in the material.

TPC removal rates from pre-diluted OME using OPR and OPB were tested in adsorption experiments. Both products were effective in removing total phenols from OME. Kinetics data were fitted to pseudo-first-order and pseudo-second-order kinetic models. Comparatively, a better fit to experimental data was obtained using the pseudo-second-order kinetic model. Adsorption isotherms were described using the Langmuir isotherm model, which provided a better fit in terms of regression coefficients compared to the Freundlich isotherm. The Gibbs free energy function was applied to investigate thermodynamics of the adsorption process. The adsorption process was found to be spontaneous and accompanied by exothermic reactions. Both OPR and OPB materials could be partially regenerated with a 70% ethanol solution.

While both OPR and OPB were effective adsorbents for the capture and removal of total phenols from OME, OPB (biochar) displayed significantly higher removal efficiency, presumably related to the higher porosity and surface area of biochar compared with raw biomass.

Considering TPC removal rates obtained at different concentrations of adsorbent material, three different approaches for the removal of phenolic compounds may come into consideration: (1) Recovery of concentrated phenolic compounds within the adsorbent matrix, allowing further valorization of the used adsorbent containing phenols as a biopesticide, at an initial dosage of 1 g·L^−1^ of OPB, could allow for or 0.0355 g·L^−1^ or 35.5% of polyphenols to be recovered, yielding a concentration of about 3.55% *w*/*w* of polyphenols captured on the adsorbent material. (2) A high dosage of adsorbent, targeting efficient depollution of the effluent and high removal rate of phenolic compounds, up to 89.1% at an initial adsorbent dosage of 20 g·L^−1^ could be considered. (3) Chemical desorption of phenolic compounds after their capture on adsorbent material could be achieved using ethanol or other solvents, allowing the recovery of phenolic compounds in soluble form for further valorization as high-value bioproducts, an option which may be also more efficient using OPB than OPR, as evidenced by a higher chemical regeneration rate.

Various studies’ findings, which are presented in Table 6, demonstrated that OPR and OPB had higher maximum adsorption capacities (q_max_) compared to recently reported adsorbents. Specifically, at pH = 10, OPR and OPB yielded q_max_ values of 21.27 mg·g^−1^ and 66.67 mg·g^−1^, respectively. If confirmed by further research, the adsorbents may also be applied for other applications, such as the removal of textile dyes from wastewater, offering effective alternatives to other adsorbents and contributing to the recycling of food and industrial waste materials.

Higher regeneration temperatures were not investigated in our study, yet may provide greater TPC removal efficiencies with regenerated adsorbent materials. Nevertheless, the low cost of the adsorbent materials used in the current study may not justify the costs of applying harsh conditions and high energy inputs for the regeneration of adsorbent materials, with addition of higher amounts of adsorbent material being a suitable alternative to the regeneration process. Hence, the use of regenerated OPR and OPB for the capture and removal of phenols may not be practical due to the significant decrease in sorption capacity after regeneration, unless the chemical regeneration processes can be coupled with the recovery of significant amounts of valuable phenolic compounds within the solvent, an option which was not yet fully investigated in our study. Hence, further studies could be carried out to investigate the optimization of regeneration methods along with the development of more effective and sustainable sorbents for the removal of phenols from industrial wastewater [40,41].

**Table 6 molecules-28-04310-t006:** Overview of phenol adsorption experiments with adsorbents based on biochar, activated carbon and olive residues.

Raw Material, Reference	Treatment of Adsorbent	Compound, Conditions	Recorded Efficiency
olive pomace Stasinakis et al. (2008) [21]	raw; dried; dried and solvent extracted; dried, solvent extracted and incompletely combusted	total phenols in olive mill wastewater	highest performance with dried, solvent extracted and incompletely combusted olive pomace
olive pomace Haydari et al. (2022) [22]	hydrogen peroxide activation	total phenols in olive mill wastewater; batch and fixed-column assays	adsorption capacity up to 789.28 mg·g^−1^ and 643.92 mg·g^−1^ at 4 g·L^−1^ concentration of phenolic compounds
olive pomace El Hanandeh et al. (2021) [23]	biochar (pyrolysis) pre-treated with FeCl_3_ prior to carbonization	aqueous solutions of phenolic compounds; raw olive mill wastewater	103.9 and 73.9 mg·g^−1^ at pH = 2 and 5, respectively, and 51.3 mg·g^−1^ using raw olive mill wastewater
olive oil solid waste (olive husk) Hamadneh et al. (2020) [24]	biochar (slow pyrolysis) followed with activation using MgCl_2_	phenol (P), PMP, PNP in aqueous solution	activation of biochar yields 1.76–2.16-fold increase of adsorption capacity compared with raw biochar
olive stone (kernels) Bohli et al. (2013) [42]	activated carbon (slow pyrolysis) 2 h 30 min, 410 °C, pre-treated with diluted H_3_PO_4_ for 9 h at 110 °C	Aqueous phenol solutions (25–300 mg·L^−1^, pH 2–9)	max. adsorption capacity 58 mg·g^−1^; highest adsorption rate at lowest pH = 2; good fit with pseudo-second-order model and Langmuir isotherm
food waste Lee et al. (2019) [37]	biochar (pyrolysis)	phenol in aqueous solution	max. adsorption capacity with biochar at max. pyrolysis temperature (700 °C); adsorption rate increases as temperature of medium increases from 15 to 35 °C
date palm Lawal et al. (2020) [43]	biochar (steam pyrolysis)	phenol in palm oil mill effluent	phenol removal rate >90% at 16–20 g/L of biochar
kenaf Cho et al. (2021) [44]	biochar (pyrolysis)	phenol in aqueous solution	max. adsorption capacity 41.1 mg/g; decreasing phenol adsorption rates as pH of medium increases from 3 to 11
date palm frond Fseha et al. (2023) [45]	biochar (pyrolysis 600 °C)	phenol in synthetic wastewater	max. phenol removal rate 64% and adsorption capacity 15.93 mg/g; optimal at pH 6, contact time 20 h (response surface methodology)
pistachio, pecan shells, wood sawdust Komnitsas and Zaharaki (2016) [46]	chemical activation of biochar (slow pyrolysis) using 1M FeCl_3_ and 1M KOH	phenol in aqueous solution	Highest adsorption capacity using biochar activated with KOH
palm kernel Hairuddin et al. (2019) [47]	magnetic biochar	phenol in wastewater	max. adsorption capacity 10.84 mg/g; pH 8 optimum
pretreated olive pomace Göktepeli et al. (2021) [48]	biochar (pyrolysis) pretreated with FeCl_3_	phenol in aqueous solution	pH 5.7 optimum; dosage of biochar 0.14 g and 60 min contact time
kiwi, cucumber, potato peels Mahmoodi et al. (2018) [49]	activated carbon	dye (methylene blue), mixtures with malachite green and rhodamine B	Endothermic, spontaneous, physical sorption following Langmuir isotherm and pseudo-second-order model
olive husk Michailof et al. (2008) [50]	activated carbon from pyrolysis 800 °C 1–3 h followed with KOH activation 800–900 °C 2–5 h, KOH:C ratio 4–6:1	mixture of phenols (caffeic acid, vanillin, vanillic acid, π-hydroxybenzoic acid and gallic acid)	Endothermic, physical and spontaneous adsorption; micropore filling may play an important role in adsorption; adsorption increases with temperature
floss silk tree waste Franco et al. (2021) [51]	activated carbon from ZnCl_2_ pretreatment followed with pyrolysis 700 °C 2 h	phenol solution 50 mg/L, adsorbent dosage 0.5–1.5 g/L	pH 7 and adsorbent dosage 0.83 g/L optimum, decreased adsorption efficiency along with increased temperature from 25 to 55 °C
olive stone Allaoui et al. (2021) [52]	crude olive stone cleaned with hexane, dried and sieved <63 μm	olive mill effluent with 0.3 g/L polyphenols, adsorbent dosage 1 g/L	adsorption efficiency decreased from 381 mg/g to 235 mg/g with increased temperature from 25 to 45 °C, indicates exothermic process
olive stone Eder et al. (2021) [53]	activated carbon from pyrolysis 600 °C 1 h followed with steam activation 900 °C 1 h	Hydroxytyrosol solution	adsorption increases with pH (range 2–8) and temperature (range 0–60 °C); based on Akaike Information Criterion, kinetics controlled via intraparticle diffusion resistance
olive stone, wood from olive tree pruning Esteves et al. (2022) [54]	activated carbon from pyrolysis 800 °C 2 h followed with CO_2_ or KOH 800 °C 4 h	Phenols (tyrosol, caffeic acid, gallic acid, protocatechuic acid, vanillic acid)	good fit with pseudo-second-order model and Langmuir isotherm, adsorption increases linearly with volume of micropores
olive stone Galiatsatou et al. (2002) [55]	activated carbon from 2-step steam activation at 850 and 800 °C of olive stone and solvent-extracted olive pulp	20% *v*/*v* diluted olive mill effluent with 1.5–2.4 g/L of polyphenols and COD 30,000–150,000 mg/L	mesoporosity may be the key factor for total phenol adsorption, microporosity effects adsorption of total organic compounds
olive pomace Abu-dalo et al. (2021) [56]	activated carbon from pyrolysis 800 °C 1 h, mixing with KOH 1 h, re-activation 800 °C 8 h, oxidation, functionalization with Cu/Cu_2_O/CuO	olive mill effluent with acid pretreatment, filtration and dilution to 124–93–62 mg/L total phenols	endothermic, spontaneous, good fit with pseudo-second-order model, adsorption increases at higher pH (range 2–11) and higher temperature (range 20–38 °C)
olive branches Vohra et al. (2022) [57]	activated carbon from pyrolysis 700 °C	phenol in aqueous solution	adsorption follows pseudo-first-order kinetics, rate constant 0.127 min^−1^
brown seaweed Rathinam et al. (2011) [58]	activated carbon from ZnCl_2_ pretreatment followed with pyrolysis 800 °C 2 h	Phenol in aqueous solution	optimal adsorption efficiency 98.31% at pH 3.0, 150 mg/L phenol, adsorbent dosage 10 g/L, time 4 h, temperature 50¨C, stirring 75 rpm
particleboard waste Girods et al. (2009) [59]	activated carbon from 3-stages: (1) 250–400 °C, (2) 800–1000 °C, (3) steam activation 800 °C 30 min	phenol solution 400 mg/L, pH 6–7	adsorption capacity up to 0.5 g/g with surface area of activated carbon 800–1300 m^2^/g
commercial activated carbon Liu et al. (2010) [60]	activated carbon fibers, 4 mm diameter (Sainuoda Co., Anshan, China)	phenol, derivatives 2-CP, 4-CP, DCP, TCP, 4-NP, DNP	best fit with Redlich-Peterson model, exothermic process as adsorption decreases at higher temperature (range 25–55 °C)
commercial activated carbon Azzam et al. (2004) [61]	activated carbon powder, Canning Chemicals, England, 50–150 μm, 800 m^2^/g	olive mill effluent treated via settling, centrifugation and filtration	at 35 °C, max. adsorption capacity reached in < 4 h, then desorption of phenols back in solution, max. adsorption 94% at 31 g/L adsorbent
commercial activated carbon Garcia-Araya et al. (2003) [62]	granular activated carbon, Hydraffin P110, Donan Carbon GmbH & Co. KG, Germany	gallic acid, p-hydroxybenzoic acid, syringic acid and their mixtures	Positive and negative interactions at low and high concentrations, respectively, max. adsorption rate 0.20–0.25 g/g after up to 150 h
commercial activated carbon Senol et al. (2016) [63]	activated carbon Sigma-Aldrich powder 100 μm, granular 600 and 1000 μm	olive mill effluent treated via centrifugation, total phenols 4821.5 g/L	max. adsorption capacity 65 mg/g and phenol removal rate 41% at 25 °C, 120 min, pH 4.4 and 30 g/L adsorbent with smaller particle size
composite material Abu-dalo et al. (2023) [64]	composite of Cu 1,4-benzene dicarboxylate metal-organic framework and granular activated carbon	olive mill effluent pH 4.0 and total phenols 440 mg/L	maximum adsorption capacity was 20 mg/g of total phenolic content (TPC) after 4 h. using 2% wt/wt of composite adsorbent
composite material Yangui and Abderrabba (2018) [65]	commercial activated carbon (Strem Chemicals) coated with milk proteins	filtrated olive mill effluent	optimal pH 7.0, 50 g/L adsorbent, max. phenol removal rate 75.4%
olive pomace (Our study)	water washed, dried 60 °C, sieved <2 mm (OPR); biochar (pyrolysis 450 °C) (OPB)	Diluted olive mill wastewater adjusted at 100 mg/L total phenols	Endothermic, spontaneous, good fit with pseudo-second-order model and Langmuir isotherm; max. adsorption capacity (OPR) 21.27 (OPB) 66.67; pH 10 optimum

## 4. Materials and Methods

### 4.1. Feedstocks

Olive pomace derived from Picholine olive variety (2022 harvest) was kindly provided by a local olive mill factory, which uses traditional batch three-phase extraction system with rotating stones (Khenifra, Morocco). Washing with distilled water was carried out several times to remove soluble organic matter from the surface of the material. The washed material was subsequently dried in the oven at 60 °C for 72 h and sieved at the desired particle size <2 mm. The product obtained was divided into two portions: the first portion, corresponding to the original product derived from raw olive pomace, was stored in glass jars and labeled OPR. The second portion was transferred to closed perforated stainless-steel boxes and pyrolyzed in a muffle furnace at 450 °C for 120 min with heating rate of 5 °C/min. The experiment was performed in three replications. Following carbonization, the resulting biochar material was also stored in glass jars and labeled OPB.

Olive mill effluent (OME) samples were collected during oil-harvesting season at the same local olive mill factory. Samples were collected in a plastic vessel prior to immediate transportation to the laboratory and subsequently transferred to a refrigerator at 4 °C for conservation. The concentration of Total Phenolic Compounds (TPC) was 22,739 ± 7615 ppm (22.739 g·L^−1^) and pH was in the range 4.9 ± 0.3. For subsequent batch adsorption experiments, OME samples were diluted about 200-fold based on initial TPC concentration, so as to achieve standardized TPC concentrations of exactly 100 ppm (0.1 g·L^−1^) in water.

### 4.2. Characterization of the Materials

Pomace-derived adsorbents PO-R and PO-B were characterized using various techniques. Surface morphology features and elemental compositions of the materials were evaluated via scanning electron microscopy as well as energy dispersive X-ray spectroscopy (SEM-EDS, JSM-5800 device, JEOL Ltd., Tokyo, Japan). The phase crystalline structure was investigated using X-ray Diffraction (XRD, Bruker D8 Advance diffractometer with Cu Kα radiation: 1.54 Å, Bruker Corporation, Billerica, MA, USA). The thermal behavior of the samples was analyzed via thermal analysis (DTA and TGA) using a thermal analyzer (DTG-60H system, Shimadzu, Kyoto, Japan) with a temperature range of 0 to 1000 °C heating at a rate 10 °C/min. Functional groups present in the materials were determined according to Fourier transform infrared (FTIR) spectra by means of a spectrophotometer (FTIR, BRUKER VERTEX 70, Billerica, MA, USA) with range 4000–400 cm^−1^ and 4 cm^−1^ resolution. Total pore volume V_Total_, pore diameter D_p_ and BET surface areas S_BET_ of both materials were estimated with Micrometrics ASAP 2020 analyzer (Micromeritics Instrument Corporation, Norcross, GA, USA) at nitrogen-adsorption desorption isotherm of 77 K.

### 4.3. Monitoring of Adsorption Experiments

The adsorption reaction was monitored with a pH meter (W&W) and high-precision balance. Folin–Ciocalteu reagent was used to evaluate total phenolic compounds (TPC) in the samples through mixing 10 mL of either diluted sample or phenol standard with 0.5 mL Folin–Ciocalteu reagent together with 1.5 mL sodium carbonate solution (200 g·L^−1^). After leaving the solution in the dark for 1 h at 20 °C, the absorbance of the solution was read at 750 nm using a spectrophotometer (Shidmadzu UV-1601, Shimadzu Corporation, Tokyo, Japan) with 1 cm quartz cells [66] against reagent blank (distilled water). Phenol standard solutions of 0, 0.1, 0.3, 0.5, 0.8 and 1 ppm were tested for calibration.

### 4.4. Implementation of Adsorption Experiments

Adsorption experiments were carried out in batch mode on pre-diluted OME samples, adjusted at exactly 100 ppm (0.1 g·L^−1^), as described in the *Feedstocks* section. The following parameters were tested in a series of experiments: initial pH of the solution (range 3–11), initial concentration of total phenolic compounds (*C*_0_: 10–100 ppm), mass of adsorbent material (*W*: 1–20 g/L) and process temperature (*T*: 298–318 K).

For this purpose, beakers (100 mL) were filled with 50 mL of different total phenolic compound (TPC) solutions and placed in a thermostated shaker. To ensure a high interfacial area of contact and improve mass transfer, the solutions were stirred at constant speed throughout the experimental period. At the end of the experimental period, treated pre-diluted OME samples were passed through filter paper (Whatman grade 42) for removal of biomass particles, and the filtrated liquid fraction was analyzed for TPC concentrations. This procedure allowed for more accurate determination of the amount of TPC removed during the adsorption process.

The pH of pre-diluted OME samples was adjusted at target values through addition of dilute H_2_SO_4_ or NaOH solutions (0.1 M).

### 4.5. Estimation and Modeling of Adsorption Parameters

The concentration of TPC retained on adsorbent material was estimated according to the following formula [67]:(2)$$Q=\frac{{(C}_{0}{-C}_{f})}{W}$$
where *C*_0_ and *C_f_* account for initial and final TPC concentrations (ppm), respectively, and *W* for adsorbent weight (g·L^−1^).

The removal rate (*R_r_*, %) of phenolic compounds was evaluated according to the following formula:(3)$$R_{r}=\left(\frac{C_{0}-C_{f}}{C_{0}}\right)\times 100$$

To ensure the data accuracy and reliability, each experiment was performed in three repetitions, and average values were calculated and reported.

Interactions between the amount of TPC absorbed and final TPC at equilibrium state were modeled according to theoretical models of Freundlich and Langmuir (Table 7) [68,69,70].

**Table 7 molecules-28-04310-t007:** Kinetic and isotherm adsorption models.

Sorption	Model	Equation	Linear Expression
Kinetic	Pseudo-first order	(4) $$q_{t}=q_{e,1P}(1-e^{-k_{1p}\cdot t})$$	(5) $$Ln\left(q_{e}-q_{t}\right)=-k_{1p}\cdot t+Lnq_{e,1P}$$
	Pseudo-second order	(6) $$q_{t}=\frac{k_{2p}\cdot q_{e,2P}^{2}\cdot t}{1+k_{2p}\cdot q_{e,2P}^{}\cdot t}$$	(7) $$\frac{1}{q_{t}}=\frac{1}{{k_{2p}\cdot q}_{e,2P}\cdot t}+\frac{1}{q_{e,2P}}$$
Isotherm	Freundlich	(8) $$q_{e}=K_{F}\cdot C_{e}^{n}$$	(9) $$Lnq_{e}=LnK_{F}+n\cdot LnC_{e}$$
	Langmuir	(10) $$q_{e}=\frac{q_{m,L}\cdot K_{L}.C_{e}}{1+K_{L}\cdot C_{e}}$$	(11) $$\frac{1}{q_{e}}=\frac{1}{q_{m,L}\cdot K_{L}\cdot C_{e}}+\frac{1}{q_{m,L}}$$

Estimated thermodynamic parameters such as Gibbs free energy (∆*G*°), enthalpy (∆*H*°), and entropy (∆*S*°) may provide indications on the spontaneity and directionality of chemical processes, according to the following equations [68]:(12)∆G°=∆H°−T∆S°
(13)$$K=\frac{Q_{e}}{C_{e}}$$
(14)$$Ln\left(K\right)=\frac{\Delta S{}^{\circ}}{R}-\frac{\Delta H{}^{\circ}}{R\cdot T}$$
where *K*, *Q_e_*, *C_e_*, *R*, and *T* account for equilibrium constant, amount of phenols adsorbed at equilibrium state (mol·L^−1^), concentration of phenols at equilibrium (mol·L^−1^), molar gas constant (8.314 J·mol^−1^·K^−1^) and absolute temperature (*K*), respectively. The slopes and intercepts of ln *K* plotted versus 1/*T* provide estimated values for ∆*H*° and ∆*S*°.

### 4.6. Regeneration Experiments

The impact of chemical and thermal regeneration on TPC removal was investigated in consecutive batch adsorption experiments, following a first batch using pre-diluted OME with TPC concentration adjusted to 100 ppm (0.1 g·L^−1^), as already described. For thermal regeneration, used adsorbent materials were thermally regenerated in an oven (Thermolyne 30400, Thermo Fisher Scientific, Waltham, MA, USA) at 100 or 200 °C for 2 h. For chemical regeneration, following a similar approach, used adsorbent materials were stirred for 2 h in 70% and 50% alcohol solutions, respectively, followed by removal of organic solvent, washing using distilled water and subsequent drying at 50 °C for 1 h. After regeneration, adsorption experiments took place under similar conditions while applying regenerated materials as adsorbents.

### 4.7. Statistical Analysis

Statistical analysis of test results was performed with the XLSTAT toolbox (2016), which is associated with the software Microsoft Excel 13. The values obtained are presented as the mean ± uncertainty at a 5% significance level for each experiment according to statistical analysis with Student’s *t*-test, with three replicates conducted for each variant.

## 5. Conclusions

The use of pomace-derived products as adsorbents for the removal of total phenols from olive mill wastewater is a promising and environmentally friendly approach to olive waste treatment. Both OPR and OPB were found to be effective adsorbents, with OPB showing higher removal and regeneration rates. The use of these materials offers a sustainable and cost-effective solution for phenol removal in wastewater treatment, while also providing potential pathways for valorization of the captured phenols as biopesticides or bioconservatives. Further research is needed to optimize and scale up the process in view of industrial application. In particular, the effects of pre-treatment processes on phenol adsorption efficiencies need to be further investigated, as well as the potential of applying concentrated OME at lower dilution rates. Additionally, recovery and valorization pathways for captured phenolic compounds should be further explored, including direct valorization of encapsulated polyphenols, which remain captured on used adsorbent material, or their valorization in soluble, desorbed form in ethanol or other solvents obtained through chemical desorption coupled with the regeneration of the adsorbents. The integration and optimization of the adsorption process, including fed-batch repeated additions of adsorbent, a continuous process and recirculation of treated effluent, should also be considered in view of achieving the required dilution rate of phenols allowing optimal operation of the adsorption process. Overall, the use of pomace-derived products as adsorbents for phenol removal from olive mill wastewater holds great potential with environmental and economic benefits, and further research in this area is warranted.

## Figures and Tables

**Figure 1 molecules-28-04310-f001:**
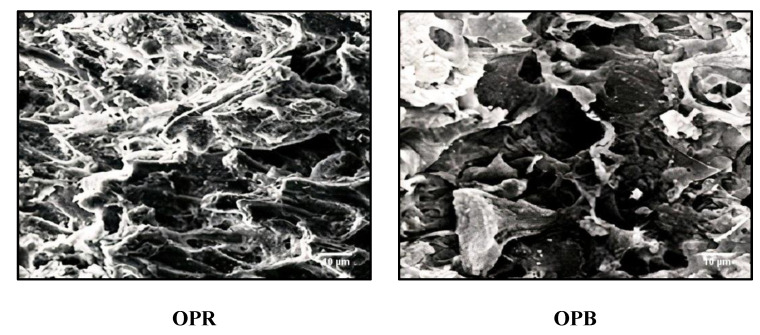
SEM images of OPR and OPB.

**Figure 2 molecules-28-04310-f002:**
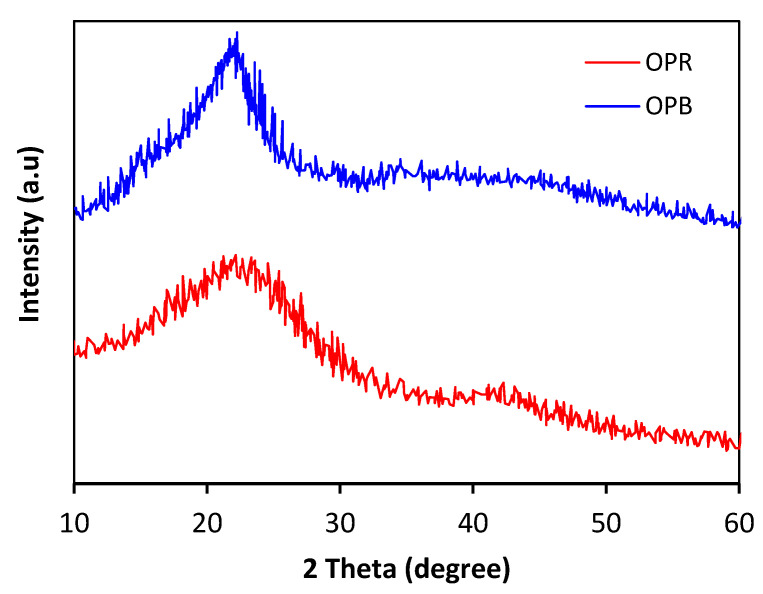
XRD spectra of OPR and OPB.

**Figure 3 molecules-28-04310-f003:**
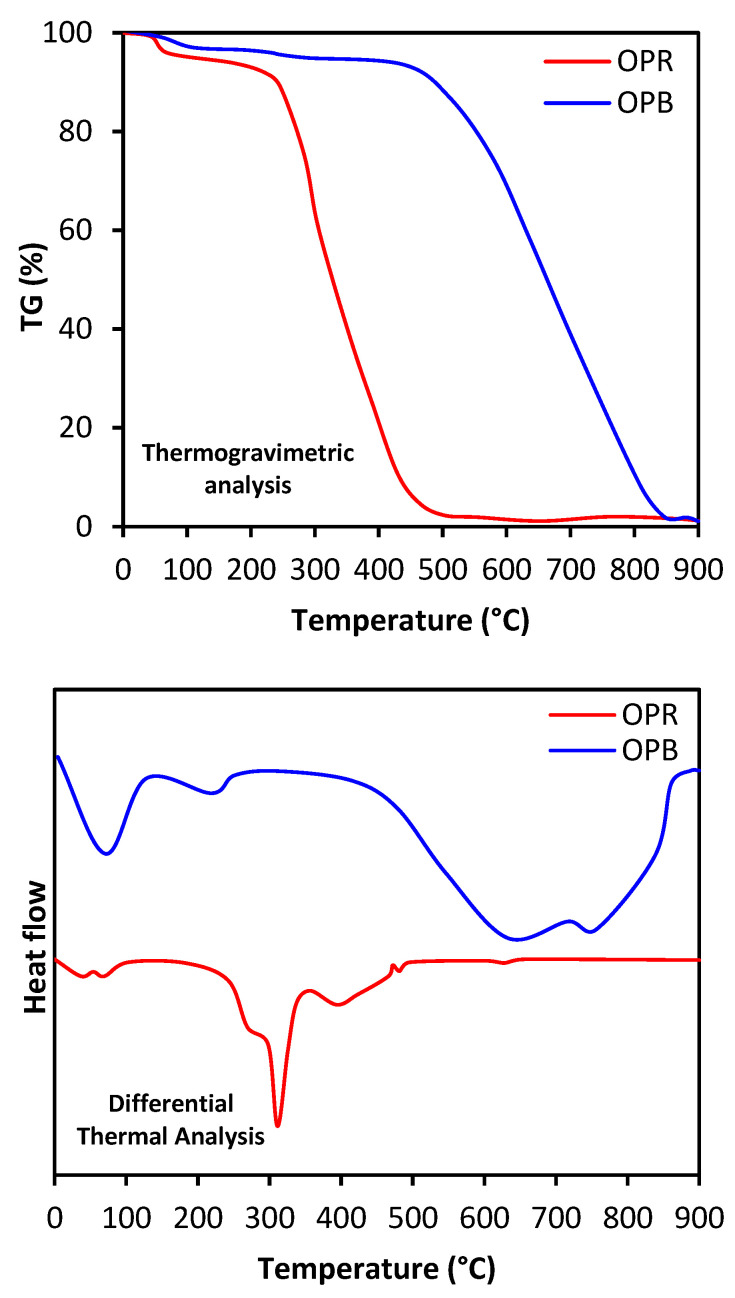
Thermal analysis of OPR and OPB via TGA-DTA.

**Figure 4 molecules-28-04310-f004:**
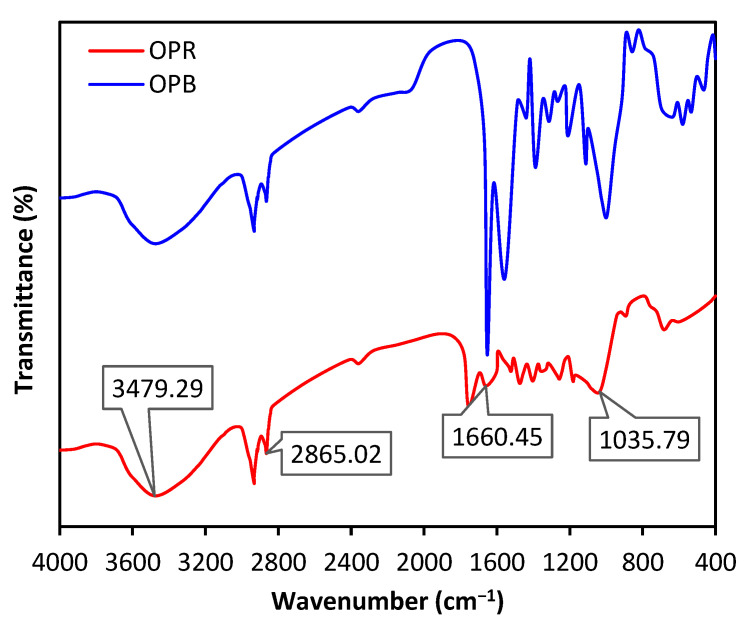
FTIR spectra OPR and OPB.

**Figure 5 molecules-28-04310-f005:**
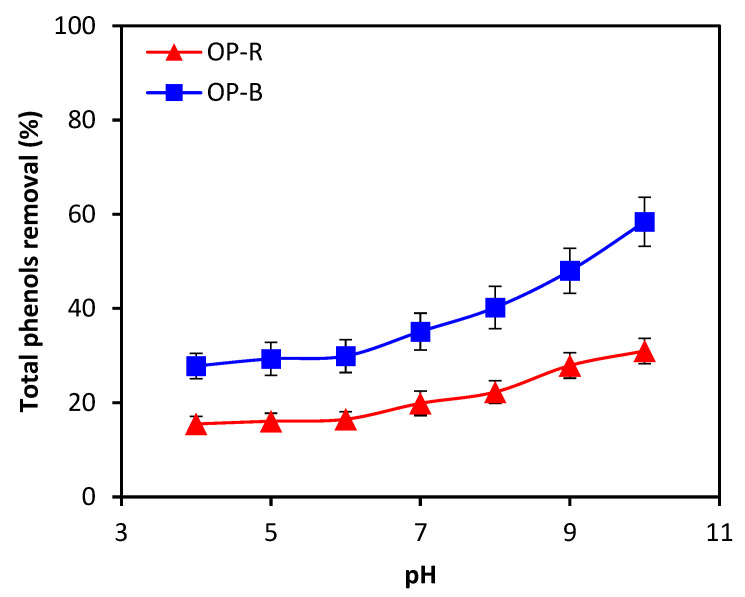
Effect of initial pH on TPC removal rate (initial TPC concentration: 0.1 g·L^−1^; dosage of adsorbent material: 2 g·L^−1^; particle size: <2 mm).

**Figure 6 molecules-28-04310-f006:**
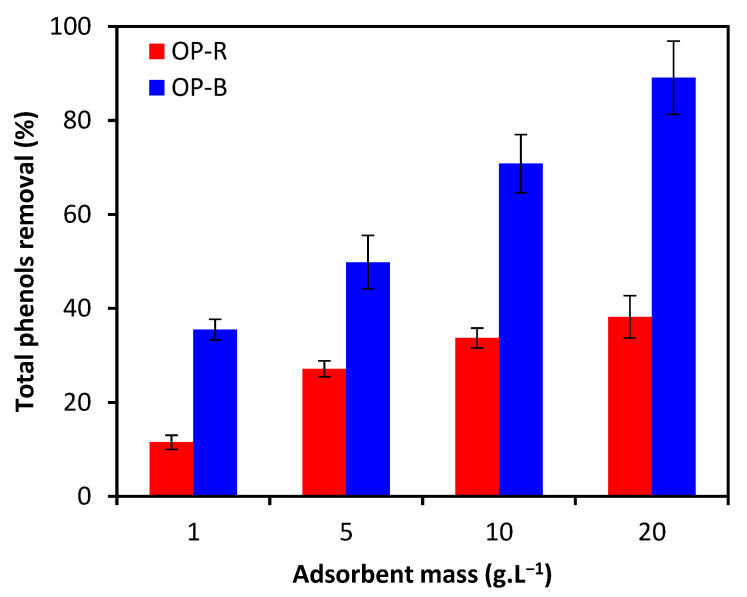
Effect of initial dosage of adsorbent material on TPC removal rate (initial TPC concentration: 0.1 g·L^−1^; particle size: <2 mm).

**Figure 7 molecules-28-04310-f007:**
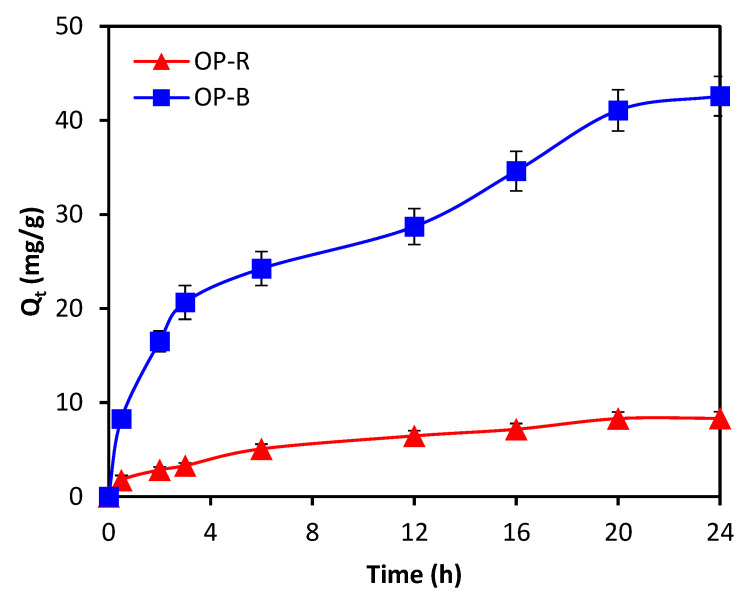
Adsorption kinetics of TPC with OPR and OPB (pH = 10, initial TPC concentration: 100 ppm; dosage of adsorbent material: 2 g·L^−1^; particle size: <2 mm).

**Figure 8 molecules-28-04310-f008:**
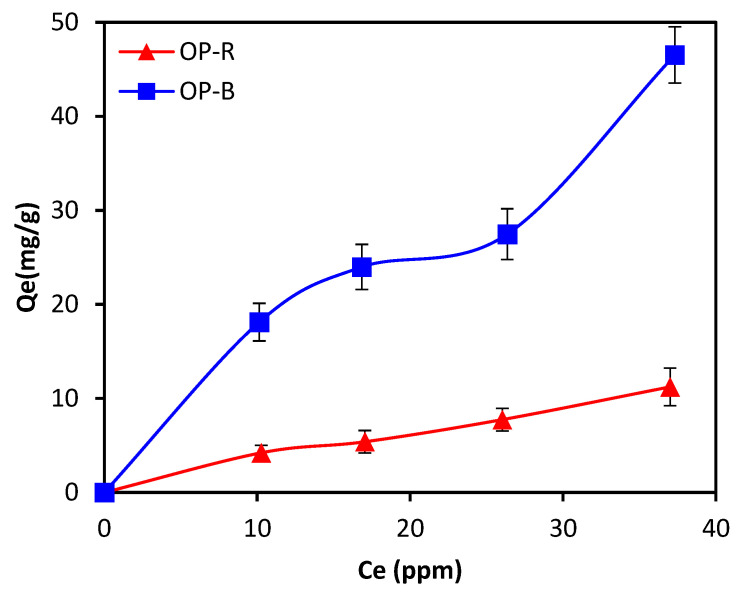
Adsorption isotherms of TPC using OPR and OPB (pH = 10, initial TPC concentration: 10–100 ppm; Adsorbent dosage: 2–20 g·L^−1^; particle size: <2 mm).

**Figure 9 molecules-28-04310-f009:**
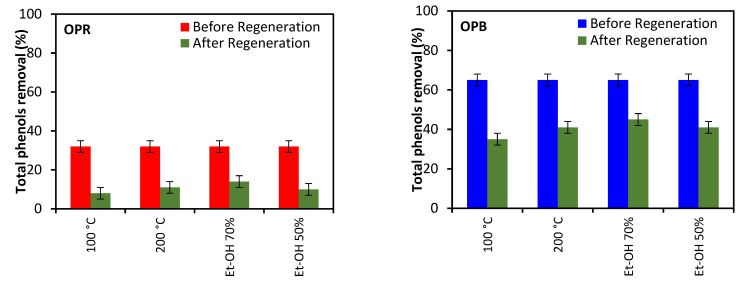
Effect of thermal and chemical regeneration on total phenols removal efficiency (initial sorbate concentration: 100 ppm; sorbent weight: 2 g; particle size: <2 mm, pH = 10).

**Table 1 molecules-28-04310-t001:** Proximate composition according to EDX analysis.

Material	C (%)	O (%)	Other Elements (%)
OPR	58.1	40.7	1.2
OPB	77.6	21.3	1.1

**Table 2 molecules-28-04310-t002:** The specific surface area of both materials.

Sample	S_BET_ (m^2^·g^−1^)	V_Total_ (cm^3^·g^−1^)	D_p_ (nm)
OPR	2.1951	0.0134	15.3725
OPB	13.6176	0.0225	3.7929

**Table 3 molecules-28-04310-t003:** Model parameters of adsorption kinetics for total phenols removal.

Model	Parameters	OPR	OPB
Pseudo-first-order	Linear expression	y = −0.117x + 1.112	y = −0.102x + 1.794
	*R* ^2^	0.848	0.737
	*k*_1*P*_ (min^−1^)	0.117	0.102
	*q_e_*_.1*P*_ (mg·g^−1^)	3.04	6.01
Pseudo-second-order	Linear expression	y = 0.225x + 0.154	y = 0.048x + 0.029
	*R* ^2^	0.902	0.970
	*k*_2*P*_ (min^−1^)	0.685	0.604
	*q_e_*_.2*P*_ (mg·g^−1^)	6.49	34.48

**Table 4 molecules-28-04310-t004:** Model parameters of adsorption isotherms for total phenols removal.

Model	Parameters	OPR	OPB
Freundlich	Linear expression	y = 0.763x − 0.173	y = 0.665x + 0.568
	*R* ^2^	0.972	0.900
	*K_F_* (L/g)	1.189	1.765
	*n_F_*	0.763	0.665
Langmuir	Linear expression	y = 2.044x + 0.047	y = 0.418x + 0.015
	*R* ^2^	0.982	0.953
	*K_L_*	0.023	0.036
	*q_m.L_* (mg·g^−1^)	21.27	66.67

**Table 5 molecules-28-04310-t005:** Thermodynamic parameters of adsorption of total phenols on the OPR and OPB.

Adsorbent	Δ*H*° (KJ·mol^−1^)	Δ*S*° (KJ·K^−1^·mol^−1^)	Δ*G*° (KJ·mol^−1^)		
			298 K	308 K	318 K
OPR	−13.15	−0.022	−6.594	−6.374	−6.154
OPB	−21.38	−0.034	−11.248	−10.908	−10.568

## Data Availability

Data is contained within the article.

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
