# Peer review of "Materials Derived from Olive Pomace as Effective Bioadsorbents for the Process of Removing Total Phenols from Oil Mill Effluents"

_molecules, 2023, doi:10.3390/molecules28114310_

Round 1

Reviewer 1 Report (Previous Reviewer 2)

The work discusses the important issue of using natural ingredients for purification. Therefore, the research topic is most justifiable. However, several elements should be improved in the article.

1. The authors compare their results with other adsorbents in Table 7. This is a good approach, but needs a broader context. In addition to sorbents based on natural ingredients, it is possible to purify with commercial, well-known sorbents of various origins, e.g. activated carbons, etc. Results should be collated for as wide a group of materials as possible so that the added value of the proposed sorbent can be shown.

2. The authors in the discussion refer to some of the results obtained related to material characteristics. I propose to elaborate by indicating whether, in the opinion of the authors, it is possible to predict the potential of a given material on the basis of a preliminary characterisation based on the proposed methods.

3. Please add a reference to the sentence “Adsorption experiments were carried out in batch mode on pre-diluted OME samples, adjusted at exactly 100 ppm (0.1 g.L-1), as described previously”.

There are linguistic errors in the work, mainly due to incorrect sentence formation.

Author Response

Dear Reviewer,

On behalf of my team, I would like to thank you for the efforts reported on our manuscript.

You will find below the comments of the proposed questions. We remain at your disposal for any suggestions to improve our manuscript.

Cordially.

  1. The authors compare their results with other adsorbents in Table 7. This is a good approach, but needs a broader context. In addition to sorbents based on natural ingredients, it is possible to purify with commercial, well-known sorbents of various origins, e.g. activated carbons, etc. Results should be collated for as wide a group of materials as possible so that the added value of the proposed sorbent can be shown.

New entries for articles related to activated carbons were added into Table 7.

  1. The authors in the discussion refer to some of the results obtained related to material characteristics. I propose to elaborate by indicating whether, in the opinion of the authors, it is possible to predict the potential of a given material on the basis of a preliminary characterisation based on the proposed methods.

The literature analysis in Table 7 has been extended regarding the link between the type and characteristics of materials and their adsorption performances.

  1. Please add a reference to the sentence “Adsorption experiments were carried out in batch mode on pre-diluted OME samples, adjusted at exactly 100 ppm (0.1 g.L-1), as described previously”.

“As described previously” was replaced with “as described in the Feedstocks section”.

Comments on the Quality of English Language

There are linguistic errors in the work, mainly due to incorrect sentence formation.

Some English language points were revised across the article, and Introduction was simplified and shortened according to the suggestion of Reviewer 2.

Reviewer 2 Report (New Reviewer)

the paper  entitled (Materials derived from olive pomace as effective bioadsorbents for the process of removing total phenols from oil mill effluents) is  a good work and sufficient effort but there are some minor revision 

1- key words do not match and sound like a general word, it should changed 

2- introduction is long, can you shorten and improve to be appropriate to the goal of paper

4- figure 2 XRD spectra the authors mentioned in (line 239, the diffractogram of the OPR materiales exhibits two shap peaks at 22.5 and 43 ) but in the figure 2 it is not like a sharp and no different from OPB , i suggest to have another look for the diffractogram and interpret the reasons clearly 

Author Response

Dear Reviewer,

On behalf of my team, I would like to thank you for the efforts reported on our manuscript.

You will find below the comments of the proposed questions.

We remain at your disposal for any suggestions to improve our manuscript.

Cordially.

Comments and Suggestions for Authors

the paper  entitled (Materials derived from olive pomace as effective bioadsorbents for the process of removing total phenols from oil mill effluents) is  a good work and sufficient effort but there are some minor revision 

1- key words do not match and sound like a general word, it should changed 

Keywords were revised.

2- introduction is long, can you shorten and improve to be appropriate to the goal of paper

Introduction was shortened.

4- figure 2 XRD spectra the authors mentioned in (line 239, the diffractogram of the OPR materiales exhibits two shap peaks at 22.5 and 43 ) but in the figure 2 it is not like a sharp and no different from OPB , i suggest to have another look for the diffractogram and interpret the reasons clearly.

Sentences for XRD spectra description were revised.

The X-ray diffraction patterns obtained from the OPR and OPB adsorbent materials indicate the absence of well-defined peaks, which means that they lack long-range order in their crystal structure, and therefore they are mostly amorphous in nature.

The presence of small crystalline phases in the amorphous matrix of the OPR material could be attributed to the cellulosic nature of olive pomace (two sharp peaks attributed to crystalline cellulose). The single crystalline phase in the diffraction pattern of the biochar OPB material suggests that it is composed primarily of carbon (an enlarged peak indicating the presence of a carbonaceous).

Round 2

Reviewer 1 Report (Previous Reviewer 2)

The authors have improved the manuscript, significantly improving its quality. I have no further comments.

This manuscript is a resubmission of an earlier submission. The following is a list of the peer review reports and author responses from that submission.

Round 1

Reviewer 1 Report

The title does not reflect the research. Authors need to change by a more adequate title.

A characterization of the adsorbents has been carried our by SEM-EDS, FTIR and XRD but the surface properties or porisity have not been analyzed. 

The introduction needs to be improved. There are some mistakes in the state of art that need to be fized.

Lines 60-62 "In particular, an efficient and cost-effective pathway is to reduce the concentration of polyphenols in OM2W is the addition of natural adsorbents, also labeled as “biosorbents”.

Lines 65-67. The mechanisms involved in the adsorption process need to be improve and well detailled. 

The most important adsorbent is activated carbon. The ion-exchange resins are based on ion exchange not in adsorption processes

Lines 79-81. authors said "The underlying principe considering the use of adsorbents is that their efficiency can be further improved through physical and chemical processing, including pyrolysis and activation for organic materials, as well as thermal treatment and addition of further minerals and ions for mineral materials".  Hoowever, pyrolysis is not performed to increase the efficiency of adsorption. The pyrolysis of organic matter is performed for other reasons. 

Discussion of the results should be improved. 

The conclusions must be re-done. They should be more specific. 

Author Response

Dear Reviewer,

We appreciate your valuable feedback and suggestions on our manuscript. Please find attached our responses to the questions you raised.

We hope that our answers address your concerns and provide the necessary clarifications. Thank you again for taking the time to review our work. 

Best regards.

Review 1:

The title does not reflect the research. Authors need to change by a more adequate title.

The title was modified.

A characterization of the adsorbents has been carried our by SEM-EDS, FTIR and XRD but the surface properties or porisity have not been analyzed. 

We have relayed the most used analysis techniques at the characterization scale such as: XRD, SEM/EDX, IRTF and GTA-DT. Porosity is an essential analysis, but we only did SEM / EDX to know the surface of the prepared materials, the analysis of surface porosity in France will take at least 30 days to do these analyzes in France (BET), which is not possible at the revision stage.

The introduction needs to be improved. There are some mistakes in the state of art that need to be fized.

The introduction has been revised and some sentences and paragraphs have been changed.

Lines 60-62 "In particular, an efficient and cost-effective pathway is to reduce the concentration of polyphenols in OM2W is the addition of natural adsorbents, also labeled as “biosorbents”.

The sentence was modified.

Lines 65-67. The mechanisms involved in the adsorption process need to be improve and well detailled. 

We have added a paragraph concerning the description of adsorption mechanisms.

The most important adsorbent is activated carbon. The ion-exchange resins are based on ion exchange not in adsorption processes.

The sentence was modified.

Lines 79-81. authors said "The underlying principe considering the use of adsorbents is that their efficiency can be further improved through physical and chemical processing, including pyrolysis and activation for organic materials, as well as thermal treatment and addition of further minerals and ions for mineral materials".  Hoowever, pyrolysis is not performed to increase the efficiency of adsorption. The pyrolysis of organic matter is performed for other reasons. 

In consideration of this comment, the sentence was modified as follows: “Adsorbents can be treated by physical and chemical processing, including pyrolysis and activation for organic materials, as well as thermal treatment and addition of further minerals and ions for mineral materials.”, and following “In particular, the conversion of organic wastes and by-products into biochar may be considered.”

Discussion of the results should be improved. 

The discussion has been revised and some sentences and paragraphs have been changed.

The conclusions must be re-done. They should be more specific. 

We rewrote the conclusion in a more concise way.

Reviewer 2 Report

Review of manuscript “Self-Sufficient Olive Waste Treatment through Removal of Total Phenols from Olive-Mill Wastewater Comparing Adsorption by Raw Olive Pomace and Olive Pomace Biochar”, presented by Fatouma Mohamed Abdoul-Latif, Ayoub Ainane, Touria Hachi, Rania Abbi, Meryem Achira, Abdelmjid Abourriche, Mathieu Brulé and Tarik Ainane.

The manuscript addresses the important issue of the removal from effluents using post-production effluent from olive pomace processing factories. Such research is in line with the global trend to reuse and recycle waste. Despite the important problem, the work needs to be corrected. The following points should be taken into account by the authors.

1. I have no tangible evidence, but the structure of the text indicates the authors' use of the ChatGPT system. Please, review the text to make it more “humane” (i.e. Abstract, .

2. In the paper the authors cited 9 of their own publications. This is not justified and should be corrected.

3. Correction of second equation is necessary.

4. The authors carried out a number of methods to evaluate the sorbents formed. In my opinion, the connection between these results and the potential adsorptive properties of the tested sorbents should be discussed in more detail, e.g., by determining which of the measured parameters indicates the potential applicability of the material and which preclude this. For example, the authors did not refer in this way to studies using the XRD method in the context of assessing the differences between the two materials, which may affect the sorption performance values used.

5. The work lacks a sorption mechanism with the material used and the contamination. It is only through this that a comprehensive analysis of the process is possible, e.g., why the yield increases with increasing pH.

6. The authors should, if possible, compare the values obtained with those for other sorbents. Only in this way is it possible to indicate the added value of the proposed solution.

7. The Authors opted for 70% alcohol in the regeneration. The decision to choose an agent should be discussed in more detail, especially as the results of the regeneration are not very satisfactory and the agent itself is not very widely used.

Author Response

Dear Reviewer,

Thank you for taking the time to review our manuscript and for providing us with valuable feedback. We appreciate your efforts to help us improve our work.

We have carefully considered all of your comments and suggestions and have made the necessary revisions to our manuscript. Please find attached a document containing our responses to your questions and comments.

Once again, thank you for your time and consideration.

Sincerely.

Review 2

  1. I have no tangible evidence, but the structure of the text indicates the authors' use of the ChatGPT system. Please, review the text to make it more “humane” (i.e. Abstract).

During the writing our team was responsible for making the manuscript, and each researcher wrote the part he did during manipulation. At the end of the drafting Prof Ainane was in charge of assembling all the parts, he only used a French to English translating tool as support, as well as "Grammarly" and "QuillBot" for the correction of grammar and spelling mistakes. As suggested by reviewer 3, a summary table was created to replace some repetitive sections of the text.

  1. In the paper the authors cited 9 of their own publications. This is not justified and should be corrected.

We removed 5 of our citations.

  1. Correction of second equation is necessary.

We corrected the 2nd equation (×100).

  1. The authors carried out a number of methods to evaluate the sorbents formed. In my opinion, the connection between these results and the potential adsorptive properties of the tested sorbents should be discussed in more detail, e.g., by determining which of the measured parameters indicates the potential applicability of the material and which preclude this. For example, the authors did not refer in this way to studies using the XRD method in the context of assessing the differences between the two materials, which may affect the sorption performance values used.

The discussion of XRD analysis was extended as follows: “According to these analyses, both products displayed porous organic textures, with OPR presumably containing a high share of cellulose, hemicellulose, and lignin, while OPB had a less crystalline, more disordered chemical structure consisting mainly of carbon. XRD is widely used for the characterization of biochar, and its main aim is to differentiate materials according to their crystalline or amorphous nature. When comparing raw biomass with biochar, the decrease in crystallinity may reveal the conversion of cellulose into amorphous carbonaceous material achieved through pyrolysis at 450°C. However, in XRD analyses of activated carbons produced through high-temperature thermal treatment of biochar, crystalline graphitic structures may appear again in the material.”

  1. The work lacks a sorption mechanism with the material used and the contamination. It is only through this that a comprehensive analysis of the process is possible, e.g., why the yield increases with increasing pH.

Increasing the pH level leads to enhanced electrostatic interactions between the polyphenols and the adsorbent. This results in stronger stability of alcohol bonds of phenols and leads to a greater amount of polyphenols being adsorbed onto the surface of biochar. It has been observed that biochar has a higher adsorption capacity towards hydroxyl ions than free ions. This theory has been supported by experimental results where the pH was increased from 4 to 10 for biochar.

  1. The authors should, if possible, compare the values obtained with those for other sorbents. Only in this way is it possible to indicate the added value of the proposed solution.

a table has been prepared which gives a comparative study of the biochar.

  1. The Authors opted for 70% alcohol in the regeneration. The decision to choose an agent should be discussed in more detail, especially as the results of the regeneration are not very satisfactory and the agent itself is not very widely used.

We conducted several tests on solvents such as acetone and methanol for the polyphenol regeneration process. However, we ultimately decided to use 50% and 70% ethanol due to both industrial and environmental considerations. This solvent choice was more cost-effective and posed fewer toxicity risks compared to the other options. Taking an industrial perspective, the use of ethanol reduces costs and increases the overall efficiency of the process. Additionally, from an environmental protection standpoint, the use of ethanol reduces the negative impact on the environment. Ultimately, the decision to use 50% and 70% ethanol was made with both practicality and sustainability in mind.

Reviewer 3 Report

1- First of all, I strongly recommend the authors provide a simple but informative graphical abstract showing the adsorption mechanism.

2- Revise the title. It is too long.

3- improve the English of the manuscript.

3-The introduction and background should be more comprehensive and provide a clear justification for the research. Consider providing a table of comparison between this work and similar studies and include the following study as well.
10.1016/j.molliq.2018.07.108

4-When a figure consists of more than one part, authors should assign the parts  (e.g., Figure 3).

5- Assign important peaks in the FTIR study.

6- The conclusion is too long and should be revised.

Author Response

Dear Reviewer,

We would like to express our gratitude for your insightful feedback, which has added significant value to our manuscript. We have carefully considered all your questions and comments, and we have provided answers and explanations where necessary.

We hope that our responses have satisfactorily addressed all your concerns and that you find our revised manuscript to be an improvement. Please accept our sincere appreciation for taking the time to review our work, and if the outcome is favorable, we extend our heartfelt gratitude.

Sincerely.

Review 3

1- First of all, I strongly recommend the authors provide a simple but informative graphical abstract showing the adsorption mechanism.

we added a graphical abstract

2- Revise the title. It is too long.

we changed the title of the manuscript

3- improve the English of the manuscript.

we have made a complete revision of the manuscript

4-The introduction and background should be more comprehensive and provide a clear justification for the research. Consider providing a table of comparison between this work and similar studies and include the following study as well.
10.1016/j.molliq.2018.07.108

we have added a summary table of previous results in the discussion part

4-When a figure consists of more than one part, authors should assign the parts  (e.g., Figure 3).

We have broken down the figures for GTA and DT to enhance clarity and facilitate easy distinction between the two materials. By making an easy comparison, we have improved the thermal analysis. This approach allows the readers to quickly and accurately understand the differences between the two materials, leading to better comprehension of the thermal properties of each. The simplified data presentation also provides a clear visualization of the key characteristics of GTA and DT, enabling more effective decision-making and informed judgments.

5- Assign important peaks in the FTIR study.

we added the labels of the important peaks of the FTIR, particularly 3430 cm-1, 2850 cm-1, 1050 cm-1 and 1600 cm-1.

6- The conclusion is too long and should be revised.

we re-wrote the conclusion in a more precise way.

Round 2

Reviewer 1 Report

Dear authors

I consider you have performed a wide review

Hovewer, there are some additional changes that are necessary in order to their publication

For example, the title "Environmental approach to the removal total phenols from effluents of olive mill wastewater using dried olive pomace and olive pomace biochar" change by "Environmental approach to the removal  OF total phenols from effluents of olive mill wastewater using dried olive pomace and olive pomace biochar"

Abstract: What is the meaning of XX%? 

I consider that it is necessary to perform a characterization of the surface area or porosity of adsorbents. You can ask for more time revision 

Additionally, the most important adsorbent material is activated carbon. However, in the introduction authors do not show that. 

Authors said "In recent years, researchers have explored new approaches to olive waste treatment, including the use of pomace-derived products as well as biochars for the removal of total phenols from OM2W". Please, include references and highligh the novelty of your research comparing to previous works. Authors said that "Pomace-derived products, such as biochar and raw material, may be effective adsorbents for the removal of total phenols from OM2W [21- 24]".

What is the novelty of your research if there are previous works that have performed the same? 

Reviewer 2 Report

The changes made have significantly improved the quality of the article. However, the authors have not addressed all of the reviewers' comments. I still hold my opinion that the adsorption mechanism should be presented in the paper. Particularly since the authors themselves have set this as a goal, see the text fragment ‘The study investigated the adsorption kinetics of total phenols by OPR and OPB, with the aim of understanding the rate and mechanism of the adsorption process’.

Reviewer 3 Report

The manuscript was amended; however, I still insist on shortening the title.